# Rheological Modeling of Bituminous Mixtures Including Polymer-Modified Binder and Fine Crumb Rubber Added through Dry Process

**DOI:** 10.3390/ma16010310

**Published:** 2022-12-29

**Authors:** Edoardo Bocci, Emiliano Prosperi, Maurizio Bocci

**Affiliations:** 1Faculty of Engineering, Università eCampus, Via Isimbardi 10, 22060 Novedrate, Italy; 2Department of Construction, Civil Engineering and Architecture, Università Politecnica delle Marche, Via Brecce Bianche, 60131 Ancona, Italy

**Keywords:** waste rubber, end-of-life tires, dry process, polymer-modified bitumen, complex modulus, Huet-Sayegh model

## Abstract

In recent years, both dry and wet processes to include waste rubber (WR) in bituminous mixtures have had little success in Italy due to technical and economic reasons. However, the necessity to recycle this waste material and the increasing interest in low-noise emitting pavement is encouraging researchers and stakeholders to explore new solutions. In this context, a hot-mix asphalt (HMA) manufactured with polymer-modified bitumen and fine WR added through a dry method has been recently developed. This paper deals with the rheological characterization and modeling of this innovative HMA, in comparison with two reference mixtures, with ordinary polymer-modified bitumen and with an asphalt rubber binder produced through a wet process. The complex modulus was measured through uniaxial cyclic compression tests at different temperatures and frequencies. The Huet-Sayegh rheological model was used to simulate the experimental data. The results showed that the time-temperature superposition principle is valid, despite the presence of polymers and WR. The Huet-Sayegh model allows a good fitting of the dynamic modulus and loss angle data to be achieved. The viscous response of the mixture including polymer-modified bitumen and rubber powder is higher than the other HMAs, which is mainly associated with the nature of the modifiers, the binder content in the mix and the less severe short-term aging.

## 1. Introduction

End-of-life tires (ELT) are one of the most voluminous waste materials worldwide. Fortunately, in Europe only a small amount of ELT currently goes to landfills, while about 95% (3.26 million tons per year) is collected and treated for energy recovery and material recycling [1]. In the second case, ELT are treated through two shredding processes that allow separating the steel, the textile fibers and the rubber, which, according to the size, is separated into [2]:chips, with dimensions between 20 and 50 mm,crumbs, with dimensions between 0.8 and 20 mm, andpowder, with sizes lower than 0.8 mm.

Among the different solutions to recycle waste rubber (WR) from ELT, the coupling of waste rubber and bituminous binder has proved to be very profitable since the first applications in the 1940s [3]. 

In the last decades, two major families of technologies for including WR into hot mix asphalt (HMA) have been identified, namely the “wet” process and the “dry” process [4]. 

In the wet process, the WR is blended with bitumen to produce a modified binder (the so-called rubberized bitumen) with improved qualities [5]. According to the processing conditions (blending temperature and time, amount and dimensions of the WR), different waste rubber modified asphalts can be obtained, typically defined as “asphalt rubber” (or “wet process-high viscosity”) and “terminal blend” (or “wet process-no agitation”) [6,7]. 

Asphalt rubber consists of a bituminous binder blended with more than 15% of WR at high temperatures (190–220 °C) for approximately 45–60 min. This allows the WR particles (that usually have dimensions of crumbs lower than 2.36 mm and contain about 25% of natural rubber from truck tires) to absorb the oily phase of asphalt and consequently swell, determining a significant increase of the binder viscosity [8,9]. However, the asphalt rubber binder is subjected to phase separation (swollen crumb rubber tends to either settle to the bottom or float near the surface of the storage tank), so it needs to be manufactured directly at the asphalt plant by equipping it with specific devices, with clear effects on the HMA cost [5,10,11,12]. 

The terminal blend is practically a polymer-modified asphalt where the modifying agent is crumb rubber. Different from asphalt rubber, in the terminal blend the WR content is lower (≤10%) and the particles are finer (passing 0.3 mm sieve) [13]. Moreover, terminal blend processing conditions are pushed to higher temperatures (200–300 °C) and shear stresses, overcoming the step of WR swelling and reaching that of WR degradation (sometimes erroneously indicated as “de-vulcanization”) [14]. This procedure is typically carried out in the refinery, and the produced binder, when properly manufactured, is stable enough (some stabilizing agents, e.g., SBS, can improve storage stability) to be handled as an ordinary asphalt binder (without requiring any agitation) [15,16]. 

Despite the noticeable improvement of the mechanical properties (resistance to fatigue, rutting, thermal and water susceptibility) and environmental benefits (recycling of waste material, reduction of the noise generated by traffic) with respect to neat asphalt, the wet process shows two main disadvantages. The first deals with the fume emissions [17]. Even if different investigations demonstrated that there are no higher risks for paving operators when dealing with rubberized asphalts [6,7], it is undeniable that, as a consequence of the high temperatures, HMA including WR generates a lot of smoke, and its smell is very pungent and irritating [18]. To solve this problem, recent studies showed that exploiting the warm mix technologies allows decreasing the paving temperatures and thus the fume emission without affecting the rubberized asphalt mixture performance [19,20,21]. The second regards the high costs, in particular in the case of asphalt rubber (that requires the plant modification for blending WR and asphalt). In Italy, there is only one plant for asphalt rubber production (so it can only provide asphalt rubber for a few HMA plants) and currently it is not always operative [22]. In such a situation, the cost of HMA with asphalt rubber is about 25% higher than HMA with polymer-modified bitumen (which in Italy only costs about 20% more than HMA with neat asphalt). For this reason, road administrations and authorities prefer investing in polymer-modified bitumen (whose mechanical performance is higher or at least equal compared to asphalt rubber), and asphalt rubber only has a small market in the field of low-noise pavements. Terminal blends are not used in Italy, as SBS-modified asphalt has a long tradition and its performance is considered to be more trustworthy. 

Some years ago, the dry process was explored as a solution to recycle WR in HMA. The method consists in adding rubber particles (with dimensions between 0.4 mm and 10 mm, so it is coarser compared to the wet process), in place of part of the aggregate, in the mixing chamber of the asphalt plant before the binder injection [23]. In this way, the interaction between WR and asphalt is limited, as it only happens during the mixing, hauling, laying and compaction of HMA until the mix temperature decreases. Therefore, in the dry process, the crumb rubber basically acts as a rubber aggregate [24,25,26]. Despite the fact that the dry process allows the recycling of a higher amount of WR (up to 3% by mix weight), the procedure was abandoned due to the poor mechanical properties and the premature failure of the HMA [27]. 

However, in recent years the dry process has come up again, thanks to the development of new technologies. One of these deals with the use of a predigested WR, named RAR (Reacted and Activated Rubber). It is an elastomeric bitumen extender, produced by hot blending a rubber granulate with selected asphalt binder and stabilizing fillers. RAR represents an intermediate between wet and dry processes, as the WR acts as an asphalt modifier but the material is cold when added to the mixing chamber of the asphalt plant [28,29,30,31,32]. 

Another innovation in the dry process regards the use of fine WR. Some studies [33,34,35] highlighted that rubber powder (RP) passing at 0.6 mm sieve can interact with asphalt binder in HMA manufactured through the dry process, allowing improving mix resistance to fatigue and rutting. Tahami et al. [36] demonstrated in the laboratory that the dry process using RP provides the best performance with a digestion time of 90–120 min and a WR content of 1–1.5% by mix weight. A Portuguese research group tested the technique in the site and observed good mechanical and surface properties of the pavement 8 years after construction [37,38]. 

In recent years, several studies have dealt with the combined use of WR and SBS to sum the benefits of the different modifiers [39,40,41,42]. In general, the contemporary use of SBS and WR allows for the increasing of the bitumen’s physical, mechanical and rheological properties [43,44], but some problems of compatibility between bitumen and the rubber/SBS modifiers were observed, which determined a low storage stability of the modified bitumen [45,46] or an excessive aging and polymer degradation [47]. However, most of the studies on the use of both SBS and WR are focused on binders produced through the wet process, while currently there are few works on the mixtures with SBS-modified bitumen and WR added through the dry process [48,49,50]. 

## 2. Materials and Methods

### 2.1. Objective and Experimental Program 

The present paper deals with a recent evolution of the dry processing, which involves a specific polymer-modified bitumen (PMB) recently engineered to include fine WR (rubber powder). The main novelty in the technology lies in the combination of polymer-modified bitumen (including SBS) and dry-processed rubber. However, this RP is not supposed to modify the bitumen in the wet process or the ordinary dry process. The RP should be digested into the bitumen only in a small amount, thus most of the rubber volume behaves as a sort of an “elastic filler”. This technology has the main scope to allow the disposal of the WR and exploit the great advantage of rubberized asphalt, i.e., the reduction of the rolling noise production. Moreover, it can be mixed and compacted at lower temperatures, thanks to the presence of WMA additive (paraffin wax) in the binder, reducing the fumes and the emissions of pollutants. The research group from the University of Bologna carried out some studies on the same technology (PMB and dry-processed fine rubber) [48,49,50]. They called it “dry-hybrid technology” and evaluated the rheological, mechanical, acoustic and environmental behavior of stone mastic asphalt (gap-graded) with and without waste rubber. However, this research did not include a comparison with other rubberized asphalt mixtures. 

For this reason, the objective of the study was the evaluation of the rheological behavior of a uniformly graded HMA made with the innovative PMB for WR dry processing, in comparison with two reference mixtures including an ordinary PMB and a wet-processed asphalt rubber binder, respectively. To this aim, uniaxial cyclic compressive tests were carried out on cylindrical specimens at different temperatures and frequencies to measure the complex modulus. The Huet-Sayegh (HS) rheological model was used to simulate the mix behavior and compare it with that of two reference mixtures: HMA with PMB and HMA with asphalt rubber. The final goal was the assessment on how the material processing or processed product properties influence the dynamic modulus and loss angle of the mixtures.

The investigated mixtures have been designated by indicating the type of bitumen (B for base bitumen, PMB for SBS-modified bitumen), the presence and type of WR (nR for no rubber, CR for crumb rubber and RP for rubber powder) and the type of rubber processing (wet or dry). The mix codes are reported in Table 1.

### 2.2. Materials and Specimen Preparation

The HMAs investigated in this research are asphalt concretes for surface layers produced in different batch plants. The PMB_nR mix included an SBS-modified bitumen containing about 5% of polymers by weight. For B + CR_wet mix, the asphalt rubber binder was manufactured in the specific production facility by mixing the bitumen and the crumb rubber at 200 °C for 60 min. The CR was obtained from cars and truck tires and was 100% passing at the 3.35 mm sieve. The base bitumen had a penetration at 25 °C of 53 × 10^−1^ mm and a softening point of 55 °C. The asphalt rubber contained 18% CR and 82% bitumen by total binder weight. After mixing, it was hauled to the asphalt plant with a tank equipped with an agitator and kept under blending until the HMA production. The PMB + RP_dry mix contained 1% by aggregate weight of rubber powder from truck tires (almost 100% passing at 1.0-mm sieve) and an SBS-modified asphalt specifically engineered for the dry processing of waste rubber. In particular, since the oily fractions of the bitumen already interacted with the SBS modifier, the rubber digestion during HMA hauling is mainly hindered. For this reason, the RP particle dimensions are comparable to the coarse fraction of fillers (the diameter ranges between 0.2 and 1 mm), and thus it does not affect the mix workability and provides elasticity, reducing the rolling noise emission. Moreover, the bitumen contains a surface-active agent, which assures a proper dispersion of the RP in the mix during HMA production in the plant, and a warm-mix asphalt (WMA) additive to increase the mix workability and allow the reducing of the production temperature at about 140–150 °C. The rubber powder was added in the mixing chamber of the batch plant during HMA production a few seconds after the addition of bitumen and before the limestone filler addition.

Table 2 shows the physical properties of the binders while Table 3 reports the particle distribution of CR and RP. 

The mix composition and aggregate gradation were determined after binder extraction by ignition at 450 °C for approximately 60 min (until constant mass was reached), according to EN 12697-39. The binder contents were 6.46%, 7.27% and 7.67% by mix weight for PMB_nR, B + CR_wet and PMB + RP_dry mixtures, respectively. The HMA mixes included basalt coarse aggregate, limestone sand and limestone filler. The aggregate gradation curves, determined according to EN 12697-2, are shown in Figure 1. The graph shows that all the mixtures complied with Italian specifications for a surface layer with a maximum aggregate size of 10 mm. Due to the acidic nature of the basalt coarse aggregate [51], an anti-stripping agent was used in all mixtures to promote the adhesion with the binder.

For each mix, two cylindrical specimens were compacted at the job site (no reheating) using a Superpave Gyratory Compactor. The compaction protocol provided 100 gyrations of 150 mm diameter and approximately 170 mm height. The specimens were then cored and sawed in the laboratory to the final dimensions of 94 mm diameter and 120 mm height. The air voids content measured on the specimens are reported in Table 4.

### 2.3. Test Protocol

The complex modulus of the specimens was determined through a servo-pneumatic testing machine. A load cell was used to measure the axial stress, while three LVDT were placed 120° apart on the middle part of the specimen (measuring base of 70 mm) to measure the axial strain. The uniaxial cyclic compressive test was carried out in control-strain configuration, applying a haversine loading with a target amplitude of 30 microstrain. Testing temperatures ranged between 10 and 50 °C with steps of 10 °C, whereas frequencies ranged between 0.1 and 20 Hz; 110 loading cycles were applied at each frequency (100 conditioning cycles plus 10 test cycles). 

The steady-state sinusoidal component of stress and strain was used to calculate the complex modulus according to Equation (1):(1)$$\;E^{*}=\frac{\sigma_{0}\exp\left[j\left(\omega t+\phi \right)\right]}{\epsilon_{0}\exp\left[j\left(\omega t\right)\right]}=\frac{\sigma_{0}}{\epsilon_{0}}\exp\left[j\phi \right]=E_{0}\exp\left[j\phi \right]=E_{1}+jE_{2}$$
where *σ*_0_ and *ϵ*_0_ are the steady-state amplitudes of stress and strain, respectively, ω = 2π*f* is the angular frequency (*f* is the testing frequency in Hz), *j* is the imaginary unit, *E*_0_ is the absolute value of *E**(also indicated as stiffness or dynamic modulus), *ϕ* is its loss (or phase) angle, and *E*_1_, *E*_2_ are its storage and loss components, respectively.

## 3. Results

### 3.1. Test Data of Stiffness Modulus and Loss Angle

Figure 2 shows the Black space (*E*_0_ as a function of *ϕ*) and the Cole-Cole diagram (*E*_2_ as a function of *E*_1_) plotting the results from all tested specimens. The Black space allowed for the observing of the interval of *E*_0_ and *ϕ*. In particular, *E*_0_ ranged between 483 MPa and 22,125 MPa for the PMB_nR mix, between 208 MPa and 9453 MPa for the B + CR_wet mix, and between 185 MPa and 17,791 MPa for the PMB + RP_dry mix. The loss angle *ϕ* ranged between 6.7° and 28.2° for PMB_nR mix, between 10.8° and 27.2° for B + CR_wet mix, and between 6.1° and 39.3° for PMB + RP_dry mix.

The graphs immediately showed that the tested materials had a frequency- and temperature-dependent behavior. However, the rheological behavior of the mixtures were different. At low temperatures (10–20 °C), the stiffness modulus *E*_0_ of PMB + RP_dry mix was intermediate between B + CR_wet (that provided the lowest *E*_0_) and PMB_nR (that had the highest *E*_0_). At high temperatures (40–50 °C), the loss angle of PMB + RP_dry mix was noticeably higher than the other HMAs. This result was probably related to the different interaction between the bitumen and the rubber particles. In particular, the presence of rubber powder in the filler fraction may have determined a higher mobility within the binder phase, entailing an increase of the loss angle. Moreover, the mix PMB + RP_dry also experienced a less severe short-term aging due to the lower HMA production temperature [52,53], which influenced the rheological behavior of the polymer-bitumen matrix. Nonetheless, the different air voids contents may partly explain the different mix behavior.

### 3.2. Application of the Huet-Sayegh Model

The time-temperature superposition principle (TTSP) can be applied if a smooth curve in the Black and in the Cole-Cole diagrams is formed by the measured *E** values of each single specimen. Even if the results plotted in Figure 2 presented a certain dispersion (PMB_nR mix) and sample-to-sample variability (PMB_nR and PMB + RP_dry mixes), a rather good overlapping between the data collected at different temperatures and frequencies was obtained, also considering that the presence of polymers in the binder generated a slight disturbance. Thus, the TTSP was considered valid. 

The master curves of the dynamic modulus *E*_0_ and loss angle *ϕ* were determined for all tested specimens at the reference temperature *T*_0_ = 20 °C (Figure 3 and Figure 4). The estimation of the temperature shift factors was carried out according to the closed form shifting algorithm [54], based on the minimization of the area between two successive isothermal curves.

The linear viscoelastic (LVE) behavior of the three mixtures was simulated using the Huet-Sayegh (HS) rheological model [55]. The model is composed (in one dimension) by a purely elastic spring (branch I) connected in parallel to two fractional derivative elements (FDE) in series with an elastic spring (branch II). The HS model is represented by the following Equation (2): (2)$$E^{*}\left(j\omega \right)=E_{e}+\frac{E_{g}-E_{e}}{1+\delta {\left(j\omega \tau \right)}^{-k}+{\left(j\omega \tau \right)}^{-h}}$$
where *E_e_* and *E_g_* are the equilibrium and glass moduli, respectively, *δ*, *h* and *k* are dimensionless model parameters (*k* < *h* < 1), and *τ* is the characteristic time. In particular, *τ* is a function of the testing temperature *T* as from Equation (3): (3)$$\tau \left(T\right)=a\left(T\right)\tau_{0}$$
where *a*(*T*) are the temperature shift factors and *τ*_0_ is the characteristic time at the reference temperature. The rheological modeling was completed with the simulation of shift factors with temperature using the Williams-Landel-Ferry (WLF) equation [56].

The master curves of *E*_0_ (Figure 3) showed that the dynamic modulus of PMB + RP_dry mix was intermediate between that of PMB_nR and B + CR_wet mixes at all reduced frequencies. In particular, it was stiffer and closer to PMB_nR at high frequencies/low temperatures, while it was less stiff and closer to WP-CR at low frequencies/high temperatures. Differently, the master curves of *ϕ* (Figure 4) showed that the loss angle of PMB + RP_dry mix was comparable to that of PMB_nR and B + CR_wet mixes at high frequencies/low temperatures (*f*_r_ > 1 × 10^−1^), but became significantly higher when moving to low frequencies/high temperatures (*f*_r_ < 1 × 10^−1^). This result denoted more marked viscous properties for the PMB + RP_dry mix compared to the others. The possible explanation for this lies in the higher mobility of the bituminous mastic due to the presence of rubber filler, whose digestion into the bitumen was hindered by the swelled SBS polymer. In addition, the presence of the WMA additive in the PMB + RP_dry binder allowed for the producing of the HMA at a lower temperature. This may have determined a mitigated effect of short-term aging and thus the preservation of the bitumen viscous features.

The HS model parameters were calculated using a least squares fitting. The graphs in Figure 3 and Figure 4 depict the HS model (lines) superimposed to the experimental data (points). It can be observed that the model was effective in simulating the *E*_0_ trend at the different reduced frequencies (Figure 5a). However, the accuracy of the HS model in predicting the loss angle was lower. In particular, the highest discrepancies between the measured and modeled loss angles (approximately 6°) corresponded to the low frequencies/high temperatures (Figure 5b), where the measured data were more scattered.

### 3.3. Analysis of Huet-Sayegh Model Parameters

The HS and WLF model parameters are reported in Table 5. It can be observed that the same values of *h* and *k* were intentionally adopted for the specimens of each family of HMA. As stated by Tschoegl et al. [57], these parameters are strictly related to the properties of the asphalt binder. This assumption is also valid for the parameters *δ* and *τ*_0_, which in fact showed a small specimen-to-specimen variability. The values of *E_g_* and *E_e_* showed a higher variability, mainly due to the air voids content of the specimens and the level of particle interlocking in the aggregate skeleton.

Figure 6 depicts the values of the parameters *k*, *h*, *δ* and *τ*_0_ for each mixture. The results showed that the value of *k* was approximately the same for the three HMAs. Differently, the parameter *h* increased when going from PMB_nR to WP-CR and finally to PMB + RP_dry. The parameters *k* and *h* represent the order of derivation of the two FDE included in the HS model and vary between 0 (purely elastic behavior) and 1 (purely viscous behavior). Therefore, the highest viscous component of the rheological behavior was obtained for PMB + RP_dry mix, and the lowest for PMB_nR mix. This result can be ascribed to the different binder content (Table 4), which may have influenced the viscous properties of the HMA. However, the rise of the viscous features can also be related to the presence of the rubber. This aspect will be more deeply investigated in the future through chemical and microscope analysis.

The values of *δ* revealed the same physical behavior. The parameter *δ* is a positive non-dimensional coefficient balancing the contribution of the first FDE in the global behavior. Thus, the values of *δ* (Figure 6c) reflected the inequality of the *k* and *h* trends (Figure 6a,b), and were higher for WP-CR and PMB + RP_dry mixes. Finally, the values of *τ*_0_ were rather low, and were comparable between the different HMA (*τ*_0_ typically varies exponentially), denoting a low relaxation time, so therefore there were good relaxation properties of the mixtures.

## 4. Discussion

In conclusion, the proposed technology can allow the combining of the high performances of the SBS-modified bitumen with the necessity of recycling WR, obtaining a mix with good rheological behavior. It has to be highlighted that the potential of the WR as a modifier is exploited only in a small way, because it likely has a reduced interaction with the hot bitumen. In fact, the previous reaction with the SBS polymer decreases the amount of available oily fractions in the bitumen [42]. This hinders the swelling of the WR particles. Moreover, the rubber powder comes in contact with the hot bitumen only for a short time during the material hauling from the plant to the construction site [48]. Thus, the interaction between WR and bitumen is further limited. 

Nonetheless, the investigated technology allows for swallowing significant amounts of WR (1% RP by mix weight was included in the studied mixture, but even higher dosages could be explored) and solving the main issues of the wet- and dry- processed asphalt rubber (storage stability, fume emissions, modifications of the HMA plant). In addition, in economic terms, the innovative PMB for RP dry processing is competitive, since within the Italian context it is noticeably cheaper than the wet-processed asphalt rubber and only slightly more expensive than ordinary PMB.

The main limitation of the present study deals with the lack of chemical and microscope analyses (e.g., Fourier Transform Infrared Spectroscopy or Scanning Electron Microscopy observations) that would allow the validation of the hypothesized interaction between rubber powder and polymer-modified bitumen in PMB + RP_dry mix. This will be addressed in future research.

## 5. Conclusions

The paper aimed at analyzing the rheological behavior of an innovative HMA manufactured with polymer-modified bitumen and fine WR (rubber powder) added through a dry process. Complex modulus testing and thermo-rheological modeling were performed on the test mixture and two reference mixtures with ordinary polymer-modified bitumen and wet-processed asphalt rubber, respectively. Based on the laboratory results, the following findings can be summarized:The LVE behavior of PMB + RP_dry is thermo- and frequency-dependent. The time-temperature superposition principle can be applied, despite the presence of polymers and WR in the binder phase.The Huet-Sayegh model can simulate the rheological response of PMB + RP_dry mix, obtaining a good fitting of the dynamic modulus and loss angle data.The dynamic modulus of PMB + RP_dry mix is lower than that of PMB_nR mix and higher than that of B + CR_wet mix at all reduced frequencies.The loss angle of PMB + RP mix reaches noticeably higher values than the other HMAs, denoting marked viscous properties.The analysis of the Huet-Sayegh model parameters confirmed the high viscous dissipation features and the good relaxation ability for the PMB + RP mixture, mainly due to the nature of the modifiers, the binder content in the mix, and the less severe short-term aging.

## Figures and Tables

**Figure 1 materials-16-00310-f001:**
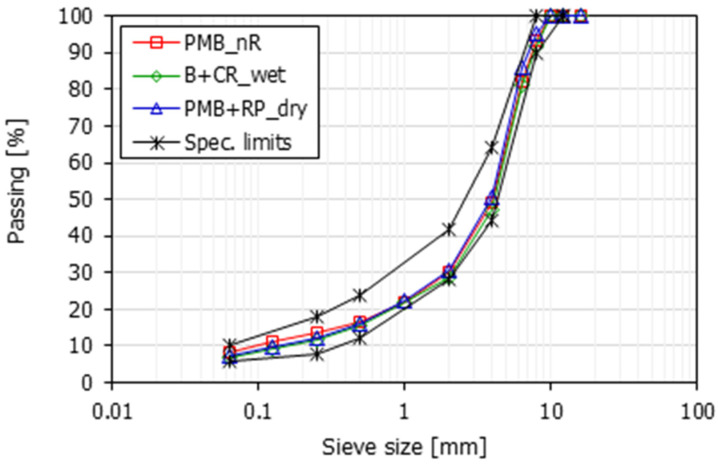
Aggregate gradations.

**Figure 2 materials-16-00310-f002:**
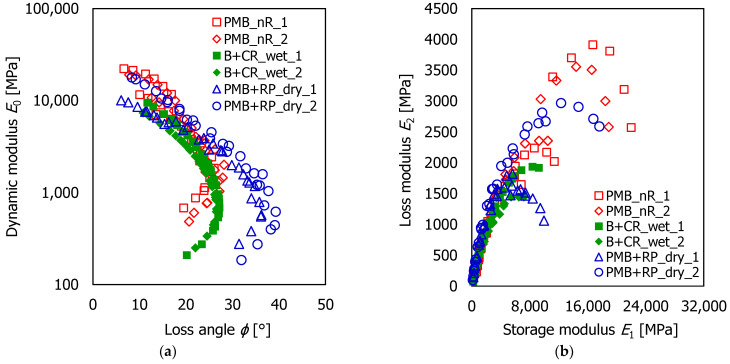
The measured values of E*: (**a**) in the Black space; (**b**) Cole-Cole diagram.

**Figure 3 materials-16-00310-f003:**
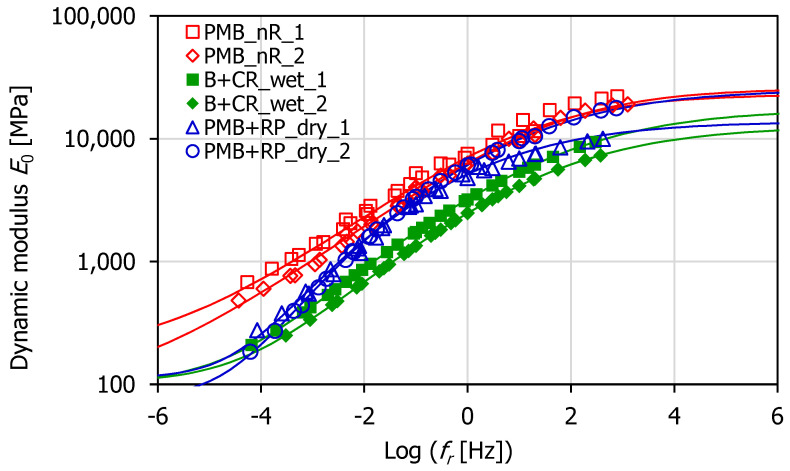
Master curves of the dynamic modulus at 20 °C.

**Figure 4 materials-16-00310-f004:**
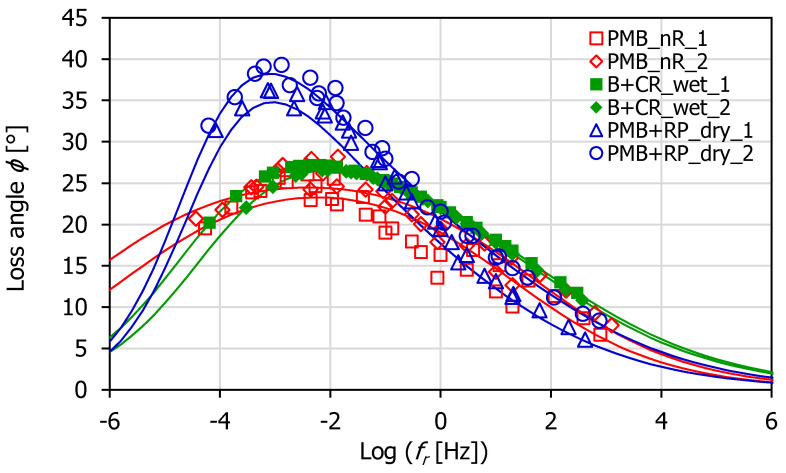
Master curves of the loss angle at 20 °C.

**Figure 5 materials-16-00310-f005:**
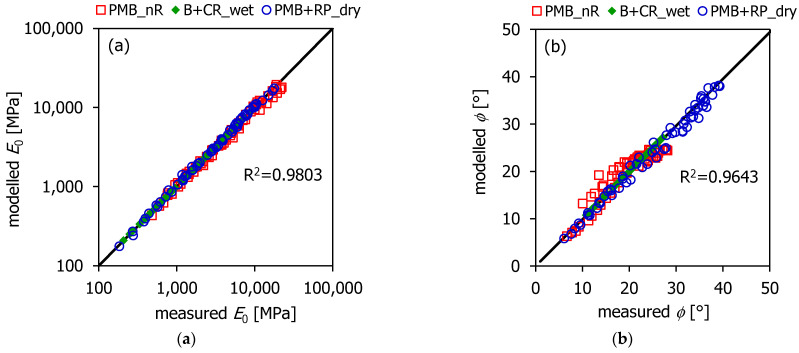
Accuracy of the HS model: (**a**) *E*_0_; (**b**) *ϕ*.

**Figure 6 materials-16-00310-f006:**
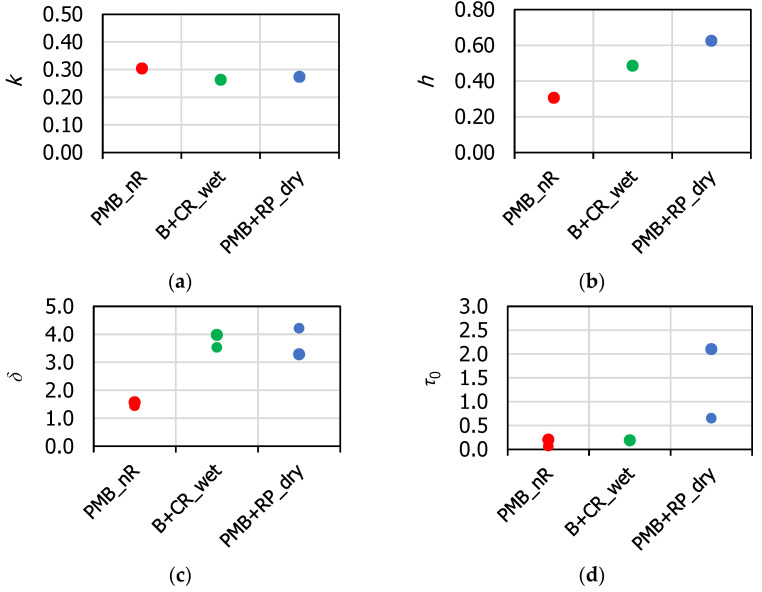
Analysis of HS model parameters: (**a**) *k*; (**b**) *h*; (**c**) *δ*; (**d**) *τ*_0_.

**Table 1 materials-16-00310-t001:** Mix codes.

Mix Code	Bitumen Type	Rubber Type	Rubber Processing
PMB_nR	SBS-modified	No rubber	-
B + CR_wet	Unmodified	Crumb rubber	Wet processing
PMB + RP_dry	SBS-modified	Rubber powder	Dry processing

**Table 2 materials-16-00310-t002:** Binder properties.

Property	Method	Mix		
		PMB_nR	B + CR_Wet	PMB + RP_Dry
Penetration @ 25 °C, 100 g/5 s (0.1 mm)	EN 1426	50–70	25–75	22–55
Softening point (°C)	EN 1427	≥70	≥54	≥ 70
Elastic recovery @ 25 °C (%)	EN 13398	≥80	–	≥75
HMA mixing temperature (°C)	–	180	180	150
Properties after RTFOT (EN 12607-1)				
Retained penetration @ 25 °C, 100 g/5 s (%)	EN 1426	≥40	≥60	≥65
Increase in softening point (°C)	EN 1427	≤5	≤12	≤8

**Table 3 materials-16-00310-t003:** Waste rubber properties.

Sieve Size [mm]	Passing [%]	
	CR	RP
3.35	100	100
2.50	97.9	100
2.36	89.7	100
2.0	55.8	100
1.4	15.3	100
1.0	3.7	99.9
0.800	0.6	91.6
0.600	0.0	49.5
0.500	0.0	37.1
0.425	0.0	27.2
0.355	0.0	16.4
0.300	0.0	10.2
0.212	0.0	3.7

**Table 4 materials-16-00310-t004:** Air voids content of the specimens.

Specimen	Air Voids Content [%]	Binder Content [%-By-Mix]
PMB_nR_1	5.02	6.46
PMB_nR_2	4.02	
B + CR_wet_1	4.33	7.27
B + CR_wet_2	6.08	
PMB + RP_dry_1	6.34	7.67
PMB + RP_dry_2	4.64	

**Table 5 materials-16-00310-t005:** HS and WLF model parameters (T_ref_ = 20 °C).

Specimen	*E_g_* [MPa]	*E_e_* [MPa]	*k* [-]	*h* [-]	*δ* [-]	*τ*_0_ [-]	*C*_1_ [-]	*C*_2_ [°C]
PMB_nR_1	23,000	175	0.304	0.306	1.568	0.203	13.7	95.2
PMB_nR_2	25,704	89			1.444	0.067	12.2	79.2
B + CR_wet_1	17,490	100	0.263	0.486	3.979	0.190	31.1	265.8
B + CR_wet_2	12,697	102			3.532	0.172	10.3	98.2
PMB + RP_dry_1	13,810	111	0.274	0.625	3.285	2.102	19.1	157.2
PMB + RP_dry_2	25,035	76			4.215	0.651	12.4	84.8

## Data Availability

Not applicable.

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
