# Peer review of "Rheological Modeling of Bituminous Mixtures Including Polymer-Modified Binder and Fine Crumb Rubber Added through Dry Process"

_materials, 2022, doi:10.3390/ma16010310_

Round 1

Reviewer 1 Report

The paper is overall well written. The study of rheological behavior of rubber based materials is useful for understanding the fundamentals, and the investigation of dry and wet- based methods on recycling waste rubbers are useful for sustainability. Moreover, the application of the huet-sayegh model is interesting but also kind of common for asphaltic mixes or asphalt rubber. 

However, there is one major issues below, regarding the impact of the study:   

  1. The paper does not include any supporting evidence on processed material properties with respective to rheological finding. For example, on page 5, the authors mentioned  “This result was probably related to the different interaction between the bitumen and the rubber particles. In particular, the presence of rubber powder in the filler fraction can have determined a higher mobility within the binder phase, entailing an increase of the loss angle. Moreover, the mix PMB+RP also experienced a less severe short- 206 term aging due to the lower HMA production temperature [40,41], which influenced the rheological properties of the polymer-bitumen matrix. Nonetheless, also the different air 2voids contents may partly explain the different mix behavior.”   The author also mentioned to validate all those findings from rheological study with chemical and microscope analysis in the future study. But it is important to at least include some evidence (e.g., FITR, or optical microscopy or SEM to show the voids and dispersion) here to support the rheological findings. Otherwise, the finding or conclusion would only be the comparison of rheological parameters but seem not be helpful for guiding material processing in the future. 

In addition, there are two minor issues and suggestions for the paper 

  1. The authors mentioned the benefits of dry process as compared to wet process, however, in the conclusion section, it is not clear that the rheological study can help draw the conclusion that dry process can be beneficial or somewhat match the wet processing. For example, the paper compared the dynamic modulus and loss angle of material from wet and dry processing, but how the dynamic modulus and loss angle affect the material processing or processed product properties? It is not clear to me yet. 
  2. In the conclusion section “In conclusion, the proposed technology can allow combining the high performances of the SBS-modified bitumen with the necessity of recycling WR, obtaining a mix with good rheological properties. It has to be highlighted that the potential of the WR as modifier is exploited only in a small part, because it likely has a reduced interaction with the hot bitumen. In fact, the previous reaction with the SBS polymer decreases the amount of available oily fractions in the bitumen, hindering the swelling of the WR particles. Moreover, the rubber powder comes in contact with the hot bitumen only for a short time, during the material hauling from the plant to the construction site.” It is not clear and not appropriate in the conclusion section neither. 

Reviewer 2 Report

This paper is aimed to assess the rheological properties of HMA containing CR; also, a model was constructed to simulate the experimental data. The manuscript is written and structured  well; however,  language editing can improve the quality and is recommended.

* Firstly, my main criticism towards this study is the lack of novelty. The usage of rubber as dry process to improve HMA performance has been tried and performed widely. Therefore, authors, in my opinion, should be able to defend the novelty of their study in a distinct section such as 'goals and objectives.'
*Authors should also summarize the research limitations and contribution to the body of literature in this section. 

* Authors are suggested  to improve the background of the study especially for previous studies conducted on HMA containing CR by dry process,Also, knowledge gap should be clarified clearer.

*Several references used in the introduction are outdated; Pls revise all references, and delete the outdated and add other recently published papers (last 2 years).

* The designation of HMA mixes used in this study should be shown in Table and showing their compositions, I think all mix codes should be revised as these codes are very confusing.

* The experiments  used is not sufficient  to evaluate the rheological properties ; therefore, authors should do more tests or focus of doing other models.

* This study mainly focused on the model simulate the experimental data. Authors should revise the Title by adding word of "Model".

* PMB+RP binder containing warm-mix asphalt (WMA); Authors should explain how this type of binder contain WMA additive and also what is the additive used. Is this type of binder common in your region?

*Authors are required to show how the dry processed CR was added into mix. was added as additive or  modifiers also, Why 8% CR was chosen? also, the same for the PMB+RP mix contained 1%?

* Pls separate the discussion before conclusion ; also, increase of the discussions of your findings.

Reviewer 3 Report

Authors have done very good work on "Rheological properties of bituminous mixtures including polymer-modified binder and fine crumb rubber added through dry process". Suggested to incorporate the below-mentioned comments in the revised manuscript.

1. Include the Literature review on the wet process and the dry process.

2. If RP is to be used as a filler, what are the basic test carried on it… does it satisfy the requirements of a filler.

3. Explain the adopted methodology in detail. So that it will understand the reader easily.

4. Give an explanation for the SB modifier, Filler activity of RP with available literature. 

5. Mention the correct specification number which is used for gradation.

6. Check Figure 1 and Table 2. I'm just confused seeing both. Simplify it if possible.

7. Remodify the discussion and conclusion part as well by considering the above suggestions.

8. Incorporate at least 5-6 references from MATERIALS.

9. Suggested to follow and incorporate the below mentioned article in the revised submission. 

A) Preethi, S., Tangadagi, B. R., Manjunatha, M., & Bharath, A. (2020). Sustainable effect of chemically treated aggregates on bond strength of bitumen. Journal of Green Engineering10, 5076-5089.

Round 2

Reviewer 1 Report

The authors provided a reasonable explanation on the objective of the research work on fundamental study on rheological behavior. The objectives, discussion, and conclusion part were properly modified. The confusion part on comparison between dry-based and wet-based processed were revised as well. Thus, I am happy to recommend to accept the manuscript at this point. 

Reviewer 2 Report

All reviewers' comments were satisfactorily addressed by authors,  thus, no further comments